# Viral Targeting of Importin Alpha-Mediated Nuclear Import to Block Innate Immunity

**DOI:** 10.3390/cells13010071

**Published:** 2023-12-29

**Authors:** Olivia A. Vogel, Jade K. Forwood, Daisy W. Leung, Gaya K. Amarasinghe, Christopher F. Basler

**Affiliations:** 1Department of Microbiology, Icahn School of Medicine at Mount Sinai, New York, NY 10029, USA; olivia.vogel@mssm.edu; 2School of Dentistry and Medical Sciences, Charles Sturt University, Wagga Wagga, NSW 2678, Australia; jforwood@csu.edu.au; 3Department of Internal Medicine, Washington University School of Medicine in St. Louis, St. Louis, MO 63110, USA; dwleung@wustl.edu; 4Department of Pathology and Immunology, Washington University School of Medicine in St. Louis, St. Louis, MO 63110, USA; gamarasinghe@wustl.edu

**Keywords:** African swine fever virus, coronavirus, Ebola virus, flavivirus, hantavirus, hepatitis B virus, human immunodeficiency virus-1, importin alpha, immune evasion, innate immunity, interferon, vaccinia virus

## Abstract

Cellular nucleocytoplasmic trafficking is mediated by the importin family of nuclear transport proteins. The well-characterized importin alpha (IMPA) and importin beta (IMPB) nuclear import pathway plays a crucial role in the innate immune response to viral infection by mediating the nuclear import of transcription factors such as IRF3, NFκB, and STAT1. The nuclear transport of these transcription factors ultimately leads to the upregulation of a wide range of antiviral genes, including IFN and IFN-stimulated genes (ISGs). To replicate efficiently in cells, viruses have developed mechanisms to block these signaling pathways. One strategy to evade host innate immune responses involves blocking the nuclear import of host antiviral transcription factors. By binding IMPA proteins, these viral proteins prevent the nuclear transport of key transcription factors and suppress the induction of antiviral gene expression. In this review, we describe examples of proteins encoded by viruses from several different families that utilize such a competitive inhibition strategy to suppress the induction of antiviral gene expression.

## 1. Introduction

Nucleocytoplasmic transport of protein cargo is mediated by a family of nuclear transport proteins known as importins, as well as other ancillary proteins such as Ran and nucleoporins [1,2,3]. The importin alpha/beta nuclear import pathway is the most well-characterized import pathway and is initiated by the recognition of a nuclear localization sequence (NLS) by one of seven importin alpha (IMPA) isoforms. The cargo-IMPA complex is bound by importin beta (IMPB) through interaction with the N-terminus of IMPA, and the trimeric complex is translocated through the nuclear pore through interactions between phenylalanine-glycine (FG) repeat regions of nucleoporin proteins and IMPB [4,5]. Following translocation across the nuclear envelope, the trimeric cargo-importin-alpha-beta complex is dissociated by RanGTP, and the importins are transported back to the cytoplasm to perform further rounds of transport (see Figure 1 for an overview of the import pathway) [6,7,8]. The initial cargo NLS recognition is mediated by one of seven importin alpha isoforms, each of which contains an N-terminal importin β-binding (IBB) domain, followed by ten armadillo (ARM)-repeat motifs [9]. Classical NLSs consist of basic amino acid clusters and are classified as being monopartite, consisting of a single cluster of basic amino acids K(K/R)X(K/R), or bipartite, which consists of two separate clusters of basic amino acids R/X(X)_10–12_KRXK [10], where X is any residue. Recognition of monopartite NLSs occurs either within the “major site”, located within ARMs 2–4, such as for SV40T-ag [11], or the minor site, located within ARMs 6–8, such as human phospholipid scramblase [12]. Bipartite NLSs, such as nucleoplasmin, occupy both the major and minor sites [11].

The seven IMPA isoforms present in humans play a pivotal role in the nuclear import of proteins, acting as adaptors that recognize and bind to NLSs on their target proteins [13,14]. These isoforms are named IMPA1 (KPNA2, or Rch1), IMPA3 (KPNA4, Qip1), IMPA4 (KPNA3), IMPA5 (KPNA1), IMPA6 (KPNA5), IMPA7 (KPNA6), and IMPA8 (KPNA7). A human IMPA2 is notably absent due to nomenclature reasons. Instead, IMPA2 refers to the murine homologue of IMPA1. The isoforms are organized into three subfamilies based on their sequence homology (Figure 2), with the SF-1 subfamily comprised of IMPA1, 8; the SF-2 subfamily comprised of IMPA3, 4; and the SF-3 subfamily comprised of IMPA5, 6, 7. These isoforms exhibit specificity for various nuclear cargoes and demonstrate unique patterns of tissue-specific expression. While each of the IMPA isoforms varies in sequence identity, ranging from 46 to 86% (Figure 2), the cargo binding site is strictly conserved, suggesting a common evolutionary origin and a fundamental mechanism of NLS recognition [15]. Despite this high conservation at the NLS binding site, the IMPA isoforms display distinct affinities for particular nuclear cargoes, underscoring the complexity and specificity of their roles in cellular function. For instance, IMPA8 is critically involved in the nuclear import of proteins essential for embryonic development, such as key transcription factors [16,17,18,19]. Structural studies have revealed the molecular basis behind IMPA3 selective binding to the W protein from Hendra virus (HeV) and Nipah virus (NiV) (see details below), and in the context of the innate immune response, IMPA3 also selectively imports NF-κB p50/p65 [20,21] (see below). The STAT family of proteins exhibits differential nuclear import by IMPA isoforms, where STAT1 and STAT3 exhibit a preference for IMPA5 [22,23,24].

## 2. Nuclear Trafficking of Cellular Transcription Factors

Inducible activation of transcription factors is a key feature of innate immune signaling pathways. This allows low basal expression and rapid upregulation of genes in response to danger signals. Several transcription factors critical for innate immunity use variations on the classical IMPA-IMPB pathway to mediate their nuclear import. NFκB is activated following the phosphorylation, ubiquitination, and degradation of the IκB proteins that bind NFκB and inhibit NFκB nuclear localization [25]. The canonical form of NFκB, which consists of p50–p65 heterodimers, interacts specifically with IMPA3 and IMPA4 for nuclear translocation [20,26]. These each possess monopartite NLSs; however, they likely bind IMPA as heterodimers that together engage both the major and minor sites [27]. This strategy may favor the nuclear import of the transcriptionally active p50-p65 heterodimer [27]. The critical antiviral transcription factor IRF3 has also been reported to selectively use IMPA3 and IMPA4, but IMPA1 has also been implicated in the nuclear transport of IRF3 and NFκB [28,29,30]. While IRF3 uses IMPA, its nuclear accumulation is triggered by its serine and threonine phosphorylation and dimerization and may reflect nuclear retention via interaction with CBP/p300 [30]. The IRF3 NLS includes critical basic residues but does not conform to the classical NLS consensus [30]. STAT1 traffics into and out of the nucleus constitutively via interactions with nucleoporins [31]. However, upon the addition of IFNs to cells, STAT1 is tyrosine-phosphorylated. This results in the formation of parallel homodimers in response to IFNγ or heterodimers with phospho-STAT2 that interact with IRF9 to form the ISGF3 transcription factor complex in response to IFNα/β or IFNλ [32,33,34,35,36,37]. This phosphorylation-mediated activation and dimerization triggers nuclear accumulation of STAT1 and STAT2 via interaction with the convex C-terminal surface of IMPα5, IMPα6, and IMPα7 [22,38,39,40,41].

The nuclear translocation of NFκB, IRF3, or STAT1 induces the transcription of a wide array of antiviral genes, including IFN and IFN-stimulated genes (ISGs). To counter these antiviral responses and promote viral replication, viruses have evolved many different strategies to inhibit these signaling pathways. Here, we will focus on viral proteins that antagonize these pathways through a competitive inhibition strategy (Table 1). In these examples, viral proteins bind components of the nuclear import machinery, including IMPA. Moreover, these viral proteins can bind the same regions of IMPA used by host cargo NLSs, impairing the nuclear import of proteins that are required for stimulating an innate immune response.

## 3. Viral Inhibition of STAT1 Nuclear Transport

### 3.1. EBOV VP24 Competitively Inhibits STAT1 Nuclear Import

EBOV is an enveloped, negative-sense RNA virus and a member of the *Filoviridae* family and *Orthoebolavirus* genus. It causes sporadic and deadly outbreaks [48]. Included among these was a very large outbreak in West Africa in 2014–2016 that was associated with approximately 28,000 cases and more than 11,000 deaths [49]. The filovirus family includes additional members of the *Orthoebolavirus* genus, Sudan virus, Bundibugyo virus (BDBV), Bombali virus, and Reston virus. Other genera include *Orthomarburgvirus*, which includes the deadly Marburg virus (MARV); *Cuevavirus*, which includes Lloviu virus (LLOV); *Dianlovirus*, which includes Měnglà virus (MLAV); and additional genera that include more distantly related viruses isolated from fish [50,51,52].

A feature of EBOV replication that promotes virulence is the efficient suppression of innate immune responses [53]. EBOV infection does not elicit robust interferon-α/β (IFNα/β) responses and renders cells unresponsive to IFNα/β-inducing stimuli such as dsRNA [54,55,56,57,58]. Additionally, treatment of EBOV-infected cells with IFNα/β or IFNγ does not result in the typical upregulation of ISGs [55,59]. The suppression of IFNα/β production can be largely attributed to the viral VP35 protein, which binds dsRNA and interacts with the host protein PACT to inhibit activation of RIG-I, blocking downstream signaling and the induction of IFNα/β expression [53]. As detailed below, the capacity to inhibit cellular responses to IFNs is due to VP24 engagement with IMPA proteins to prevent the function of STAT1, a transcription factor central to IFNα/β, λ, and γ signaling.

VP24, when expressed alone, suppresses IFNα/β or IFNγ-induced gene expression and prevents the establishment of an IFNβ-induced antiviral state in cells [60]. This VP24 function likely explains the failure of EBOV-infected cells to respond to IFNs. When IFN is added to VP24-expressing cells, STAT1 tyrosine phosphorylation remains intact. However, tyrosine-phosphorylated STAT1 did not accumulate in the nucleus of VP24-transfected or EBOV-infected cells. The effect was specific to STAT1, because VP24 expression did not inhibit nuclear translocation of IRF3. Examination of the STAT1 interaction with IMPA5 (KPNA1) demonstrated that the interaction, as assessed by co-immunoprecipitation, could be triggered by the addition of IFN to cells. However, when VP24 was present, the IFN-induced STAT1-IMPA5 interaction was abrogated [60]. Mapping studies revealed overlapping IMPA5 binding sites for tyrosine-phosphorylated STAT1 and VP24. Inhibition of tyrosine-phosphorylated STAT1 binding to IMPA5 depended on the amount of VP24 present [61]. Further, VP24 and STAT1 both interact with IMPA5, 6, and 7 (KPNA1, 5, and 6), which comprise the SF-3 IMPA subfamily (also referred to as the nucleoprotein interactor 1 (NPI-1)) subfamily in some studies [39,41]. Whereas the STAT1 interaction with IMPAs depends on activation by IFN, the VP24-IMPA interaction does not require an activating stimulus [60,61]. Point mutations in VP24 that disrupted IMPA5 interaction and which later structural studies demonstrated lie close to key VP24-IMPA contacts also abrogated inhibition of IFNβ signaling [41,62]. A different study demonstrated direct interaction between VP24 and a truncated version of unphosphorylated STAT1, suggesting an additional mechanism of inhibition [63]. However, while VP24 interactions with IMPA have been supported by several cell-based proteomics studies, interactions with endogenous STAT1 have not been reported [64,65,66]. This, combined with the loss of IFN inhibition by IMPA-binding mutants, suggests a model where VP24 competes with tyrosine-phosphorylated STAT1 binding to IMPA5, 6, and 7 to prevent STAT1 nuclear import and to prevent ISGs from being expressed.

Highly detailed insight into VP24-IMPA interaction and VP24 suppression of IFN signaling was provided by biophysical and structural studies that demonstrated direct interaction of purified VP24 with the C-terminal ARMs 7–10 of IMPA6 [41,63]. The X-ray crystallography structure of the complex demonstrated that three clusters of amino acids on VP24 make extensive contacts with the IMPA C-terminus [41]. Mutations in these clusters disrupted VP24-IMPA interaction, inhibition of IFN signaling, and STAT1 nuclear accumulation [41]. Together, this underscored the importance of the VP24-IMPA interface for the inhibition of IFN signaling. Furthermore, it was demonstrated that, like tyrosine-phosphorylated STAT1, VP24 interacts with a non-classical NLS (ncNLS) binding site found in ARMS 8–10 of IMPA6 [41]. The significance of this ncNLS was examined in additional co-immunoprecipitation experiments demonstrating that VP24 did not inhibit IMPA5 interaction with the deleted in breast cancer 1 (DBC1) protein, which possesses a cNLS [41]. This suggests that by utilizing an ncNLS binding site on IMPA, VP24 is able to target STAT1 nuclear import and STAT1-dependent gene expression without impairing other IMPA-cNLS interactions. The retention of cNLS binding may allow cNLS-mediated nuclear trafficking pathways to remain intact (Figure 3). While the VP24-IMPA interaction appears designed to block STAT1 nuclear import, VP24 also impairs the interaction of heterogenous nuclear ribonucleoprotein complex C1/C2 (hnRNP C1/C2) with IMPA5 and decreases its nuclear accumulation [67]. This suggests that VP24 affects the nuclear transport of other host proteins that utilize an ncNLS binding site on SF-3 IMPA proteins. How broadly VP24 affects nuclear import remains to be fully elucidated.

Sequence analysis revealed that among the three clusters of VP24 residues that comprise the IMPA binding interface, clusters 1 and 3 were highly conserved among viruses in the *Orthoebolavirus* genus [41]. Furthermore, the ability of VP24 to interact with IMPA and inhibit STAT1-dependent gene expression is conserved within the *Orthoebolavirus* genus but not the *Orthomarburgvirus* genus [41,61,68]. Unlike EBOV, Marburg virus (MARV) VP24 does not interact with IMPA to inhibit IFN signaling [41,68,69]. Instead, MARV VP24 induces cytoprotective antioxidant responses through an interaction with the host protein Keap1 [70,71]. Interestingly, when looking beyond these two filovirus genera, the VP24-IMPA interaction is conserved in LLOV but not MLAV [72,73].

While binding of VP24 to the SF-3 IMPA subfamily is conserved in the *Orthoebolavirus* genus, the strength of interaction can vary between viruses [68]. For example, BDBV VP24 has a lower binding affinity for IMPA compared to EBOV VP24 [68]. This lower binding affinity was also correlated with reduced inhibition of IFN signaling [68]. The interaction between VP24 and IMPA also increases the stability of VP24 [68]. Experiments examining VP24 expression after cycloheximide treatment demonstrated that overexpression of SF-3 subfamily IMPA proteins prolonged VP24 half-life [68]. Likewise, VP24-IMPA interaction mutants exhibited reduced half-lives when compared to the VP24 wildtype [68]. Providing evidence that the affinity of VP24-IMPA interaction influences the outcome of infection, swapping IMPA binding residues of BDBV VP24 into EBOV VP24 attenuated virus growth by both IFN-dependent and IFN-independent mechanisms [74].

### 3.2. HBV Polymerase Inhibits STAT1 and STAT2 Nuclear Translocation

Hepatitis B virus (HBV) is a member of the *Hepadnaviradea* family and possesses a partially double-stranded, circular DNA genome [75]. HBV is transmitted primarily through infected blood and bodily fluids, leading to acute, self-limiting, or chronic HBV infection (CHB) [75]. HBV is efficient at suppressing IFN signaling, with one mechanism being the suppression of STAT1 nuclear localization by HBV polymerase (Pol) [42,76,77]. To examine how Pol affects STAT1-dependent gene expression, a HBV viral replicon was used to examine IFN-α-induced ISRE promoter-driven luciferase activity [42]. ISRE reporter suppression was compared after transfection of a wildtype HBV replicon and a Pol-null-HBV replicon [42]. It was observed that the Pol-null-HBV replicon was unable to suppress ISRE reporter activity to the same degree as the wildtype HBV replicon in Huh7 cells, suggesting the Pol expression suppresses IFN-α-induced ISRE promoter activity [42]. Additionally, C57BL/6 mice that were hydrodynamically injected with HBV or Pol plasmids exhibited reduced Mx1, STAT1, MyD88, and TAP-1 expression compared to empty vector control-injected mice [42]. Together, this confirmed that Pol expression suppresses IFN-α-induced gene expression [42]. Focusing on STAT1 and STAT2 activation, it was observed that Pol overexpression suppressed STAT1 Ser727 phosphorylation, which is important for STAT1 transcriptional activity [78,79]. In contrast, Pol expression did not inhibit IFN-α-induced STAT1 and STAT2 tyrosine phosphorylation [42]. Furthermore, Pol expression did not prevent STAT1-STAT2 heterodimerization induced by IFN-α, as assessed by co-immunoprecipitation experiments [42]. Focusing first on STAT1 Ser727 phosphorylation, it was determined that Pol interacted with the catalytic domain of the kinase PKC-δ, which helps to regulate STAT1 Ser727 phosphorylation [42]. It was also found that Pol expression inhibited IFN-α-induced STAT1 and PKC-δ interaction in co-immunoprecipitation experiments, suggesting that Pol and PKC-δ interaction may prevent STAT1 Ser727 phosphorylation [42]. Through immunofluorescence assays and subcellular fractionation assays, it was confirmed that Pol suppresses STAT1 and STAT2 nuclear accumulation in Huh7 and HepAD38 hepatocyte cell lines [42]. Using biopsies from CHB patients, it was also observed that STAT1 and STAT2 nuclear accumulation was reduced in cells infected with HBV, as determined by immunofluorescence [42]. Through co-immunoprecipitation experiments, it was demonstrated that Pol expression led to reduced interaction between IMPA5, STAT1, and STAT2 after treatment with IFN-α [42]. This was also correlated with impaired colocalization between STAT2 and IMPA5 in immunofluorescence experiments [42]. Additional co-immunoprecipitation experiments demonstrated that Pol interacts with the same region on IMPA5 that STAT1 binds, suggesting that Pol can bind IMPA5 to competitively inhibit STAT interaction and prevent STAT1/2 nuclear translocation [42]. Additional co-immunoprecipitation experiments with Pol truncation mutants demonstrated that different Pol domains are required to inhibit PKC-δ-STAT and IMPA5-STAT interactions, with the Pol terminal protein domain binding IMPA5 and the Pol RNase H domain binding PKC-δ [42]. These results demonstrate that HBV Pol utilizes different domains to competitively inhibit the interaction between STATs and host proteins, ensuring efficient suppression of STAT-mediated gene expression.

### 3.3. SARS-CoV and SARS-CoV-2 ORF6 Interaction with IMPA and Inhibition of STAT1 Nuclear Translocation

SARS-CoV belongs to the *Sarbecovirus* subgenus within the *Betacoronavirus* genus of the family *Coronaviridae*. It emerged from an animal source and caused disease in approximately 8000 people in 2002–2003. These infections were associated with severe respiratory disease and an approximately 9.5 percent case fatality rate [80]. In a variation on the competition mechanism, the protein encoded by ORF6 of SARS coronavirus (SARS-CoV) was demonstrated to block IFN-induced STAT1 nuclear accumulation through interaction with IMPA [81]. Interestingly, ORF6 binds IMPA1 and indirectly impacts STAT1 nuclear import [43]. ORF6 localizes to the endoplasmic reticulum (ER) and Golgi membrane. This recruits IMPA1 and IMPB to these membranes. When an IMPB-binding domain mutant of IMPA was examined, it bound ORF6. However, IMPB was absent from the complex, and STAT1 nuclear import was not impaired. This suggests a model where ORF6, via its interaction with IMPA, sequesters IMPB such that IMPB is not available to mediate the nuclear import of STAT1 [43]. Such a mechanism would be expected to have broad effects on IMPA/IMPB nuclear import. Consistent with this, a combination of transcriptomics, proteomics, and chromatin immunoprecipitation assays comparing SARS-CoV and a SARS-CoV lacking ORF6 demonstrated that ORF6 decreases the activity of many transcription factors that rely on IMPA for nuclear import [82]. Interestingly, SARS-CoV-2, a related betacoronavirus and the cause of the COVID-19 pandemic, also encodes an ORF6 [80]. This ORF6 also interacts with IMPA1 as well as IMPA5 [83,84]. However, while SARS-CoV-2 ORF6 blocks STAT1 and STAT2 nuclear import, this has been attributed to ORF6 interactions with the Nup98-Rae1 complex at the nuclear pore, such that IMPA/IMPB docking at the nuclear pore is prevented [83]. As might be expected of such a mechanism, the interaction of ORF6 with the Nup98-Rae1 complex has broader effects on nuclear trafficking, including inhibition of mRNA nuclear export and impacts on nuclear import of interferon regulatory factors and proteins other than STAT1 [84,85,86,87,88].

## 4. Viral Inhibition of IRF3 and NFκB Nuclear Transport

### 4.1. MERS-CoV ORF4b

A number of viruses competitively block the nuclear import of IRF3 and/or NFκB. The MERS-CoV accessory protein ORF4b is a well-characterized example of such a viral protein. First identified in Saudi Arabia in 2012, MERS-CoV belongs to the *Merbecovirus* subgenus within the *Betacoronavirus* genus [89]. Endemic to dromedary camels, MERS-CoV has been shown to cause sporadic outbreaks, and a vaccine is not currently available [90]. ORF4b possesses an N-terminal nuclear localization domain and a C-terminal 2′, 5′ phosphodiesterase (PDE) domain that can inhibit activation of the antiviral oligoadenylate synthetase (OAS)-RNase L pathway [91,92]. In a screen examining MERS-CoV structural and accessory proteins for IFN antagonism, ORF4b was identified as a potential IFN antagonist along with the ORF4a, ORF5, and membrane (M) proteins [93]. Unlike the other IFN antagonists identified by the screen, ORF4b was predominantly nuclear in localization [93]. Furthermore, ORF4b expression was correlated with reduced nuclear localization of GFP-IRF3, suggesting ORF4b could target IRF3 activation [93]. Additional studies demonstrated that ORF4b could also inhibit activation of NFκB-dependent gene expression [94]. ORF4b nuclear localization and inhibition of IFN-β and NFκB-mediated gene expression were also found to be conserved in the closely related MERS-like bat HKU5 and HKU4 viruses, suggesting that ORF4b-mediated innate immune antagonism may be conserved among other viruses in the *Merbecovirus* subgenus [94]. The role of ORF4b as an innate immune antagonist was further confirmed in studies utilizing a MERS-CoV lacking expression of ORF4b [26]. Compared to wildtype MERS-CoV-infected cells, cells infected with the ORF4b deletion virus exhibited elevated expression of inflammatory cytokines [26]. Using a mouse-adapted MERS-CoV, deletion of ORF4b led to viral attenuation, improved survival, and reduced lung pathology in mice expressing the MERS-CoV receptor, dipeptidyl peptidase 4 (DPP4) [95]. Together, this demonstrates that ORF4b is an important virulence factor in MERS-CoV infection [26,95].

Examination of the ORF4b sequence revealed a potential bipartite NLS, mutation of which resulted in impaired ORF4b nuclear import in overexpression studies or experiments using ORF4b NLS mutant viruses [26,94]. Co-immunoprecipitation experiments demonstrated that ORF4b can interact with IMPA proteins, and mutations in the putative NLS led to impaired IMPA interaction [26,44]. During infection, it was observed that ORF4b nuclear localization correlated with impaired nuclear import for the p65 subunit of NFκB, as assessed by subcellular fractionation experiments and immunofluorescence microscopy [26]. Infection with ORF4b NLS mutant viruses impaired nuclear import of ORF4b and restored NFκB responses [26]. ORF4b expression did not impact IκBα degradation or p65 expression, suggesting that ORF4b acted downstream of these events to inhibit the expression of inflammatory cytokines [26]. Co-immunoprecipitation experiments also revealed that ORF4b-IMPA3 interaction impeded IMPA-p65 interaction in wildtype MERS-CoV-infected cells [26]. However, p65 interaction with IMPA3 was restored in cells infected with an ORF4b deletion mutant or an NLS mutant MERS-CoV [26]. Together, this established that ORF4b could interact with IMPA to inhibit IRF3 and NFκB nuclear import.

High-resolution crystal structures of the ORF4b NLS bound to IMPA2 and IMPA3 provided additional insight into the nature of the ORF4b-IMPA interaction (Figure 4) [44]. IMPA2, the mouse homolog of human IMPA1, was chosen because it is the most structurally studied IMPA isoform in the Protein Data Bank (PBD), while human IMPA3 was used based on prior studies demonstrating that, among the IMPA proteins tested, ORF4b bound IMPA3 with the highest affinity [26,44]. ORF4b is bound at both the major and minor sites of IMPA, with Arg^24^ and Arg^33^ of ORF4b serving as key binding determinants at the minor and major sites, respectively [44]. Mutation of these key residues led to reduced IMPA binding in both fluorescence polarization and co-immunoprecipitation assays, with the Arg^33^ mutant having the strongest impact on binding [44]. These mutants also demonstrated reduced nuclear localization and an impaired ability to inhibit IFNβ and NFκB promoter activity, further highlighting the importance of the ORF4b-IMPA interaction for MERS-CoV innate immune antagonism [44]. Fluorescence polarization and co-immunoprecipitation assays also demonstrated that ORF4b did not have an IMPA isoform preference and was able to bind members of all the IMPA subfamilies [44]. Therefore, ORF4b has the potential to inhibit the nuclear translocation of a wider range of transcription factors, suggesting a broader suppression of antiviral signaling. Future studies are required to determine the range of transcription factors targeted by ORF4b.

Interestingly, it was observed that MERS-CoV ORF4b does not have a canonical Lys residue at the P2 position in the major cNLS binding site [44]. Instead, Arg^33^ was found positioned in this region [44]. Crystal structures of ORF4b NLS mutant peptides bound to IMPA2 revealed that mutation of Arg^24^ to alanine led to binding only at the major site instead of the major and minor sites as observed with the wildtype NLS [44]. This mutant also exhibited a shift in binding at the major site, with a canonical lysine occupying the P2 position instead of the previously identified Arg^33^ [44]. A similar shift was observed when Arg^33^ was mutated to alanine [44]. To determine whether this unusual binding pattern was conserved, the crystal structure of the HKU5 ORF4b NLS peptide bound to IMPA2 was examined. Unlike the MERS ORF4b NLS, the HKU5 NLS binds IMPA2 with a canonical Lys at the P2 position of the major binding site [44]. Further examination is required to understand the importance of this unique binding mechanism. This sequence flexibility within the ORF4b NLS may allow evolution of the NLS region without loss of IMPA major site interaction.

To better understand the mechanism by which ORF4b inhibits NFκB nuclear import, a crystal structure of the NFκB p50 NLS bound to IMPA2 and IMPA3 was examined [44]. This structure revealed that the p50 NLS bound at the major site of IMPA, supporting results from previous co-immunoprecipitation experiments using IMPA3 major and minor site mutants [20,44]. Structural comparison revealed that the ORF4b NLS binding site overlapped with the p50 NLS binding site [44]. A recent study also demonstrated that the p50 and p65 subunits of NFκB bind IMPA3 as a heterodimer [27]. In alignment with prior studies, they confirmed that the p50 NLS binds at the IMPA3 major site and also demonstrated that the p65 NLS binds at the minor site of IMPA3 [27]. Together, these studies fit within the competitive inhibition model for ORF4b antagonism. By binding to both the major and minor binding sites of IMPA, the ORF4b bipartite NLS could effectively compete with both p50 and p65 binding to inhibit NFκB nuclear import.

### 4.2. Hantaan Virus Nucleocapsid (N) Protein

Hantaan virus (HTNV) is a segmented negative-sense RNA virus that belongs to the *Orthohantavirus* genus and *Bunyaviridae* family. Infection with pathogenic “Old World” hantaviruses, like HTNV, can cause hemorrhagic fever with renal syndrome (HFRS) [96]. Pathogenic “New World” hantaviruses, such as Andes virus (ANDV) or Sin Nombre virus (SNV), can cause hantavirus cardiopulmonary syndrome (HPS) [96]. Both types of pathogenic hantaviruses are effective at suppressing the induction of IFNβ expression during infection. For example, one of the two surface glycoproteins, Gn, utilizes its cytoplasmic tail to antagonize RIG-I/MAVS-mediated signaling [97,98,99]. Additional hantavirus proteins implicated in the suppression of innate immune signaling include the N protein, the glycoprotein precursor protein, and a nonstructural protein [100,101]. Among these viral proteins, the HTNV N protein has also been shown to interact with IMPA proteins to inhibit NFκB-dependent gene expression [102]. In a study aimed at identifying HTNV viral proteins capable of antagonizing TNFα-mediated signaling, it was observed that the HTNV S segment, encoding the N protein, could inhibit NFκB promoter activity following treatment with TNFα [102]. Further examination revealed that N expression did not impact the expression of the p50 and p65 subunits of NFκB or the degradation of IκBα, suggesting that HTNV N functions downstream of NFκB activation. Co-immunoprecipitation experiments determined that while HTNV did not interact with p65 or p50, it could interact with IMPA1, 2, and 3 but not IMPA4 [102] (It is possible that IMPA1–4 in this study corresponds to human KPNA1–4, i.e., IMPA5, IMPA1, IMPA4, and IMPA3). Given that NFκB nuclear transport can be mediated by IMPA3 and IMPA4, this suggests that HTNV N could competitively inhibit NFκB nuclear translocation [20,30]. Indeed, HNTV N expression was correlated with reduced expression of p65 in the nucleus after TNFα treatment in immunofluorescence and nuclear fractionation experiments. Other hantavirus N proteins, including those from HFRS-associated Seoul and Dobrava viruses, have also been shown to interact with IMPA1, 2, and 3 and inhibit p65 nuclear transport [103]. In contrast, N proteins from the HPS-causing ANDV and SNV were unable to suppress p65 nuclear transport or inhibit NFκB activation induced by TNFα [103]. While not explored in these studies, IRF3 has also been shown to translocate to the nucleus via an interaction with IMPA3 and IMPA4, suggesting that HTNV N could potentially inhibit IRF3 nuclear translocation to broadly disrupt antiviral gene expression. Further research is required to determine the binding mechanism of HTNV N with IMPA and how this impacts NFκB and IRF3 nuclear import.

### 4.3. Japanese Encephalitis Virus Nonstructural Protein 5 (NS5)

Another virus that can inhibit IRF3 and NFκB nuclear import is the mosquito-borne Japanese encephalitis virus (JEV) of the *Flaviviridae* family [104]. Infection with JEV can cause viral encephalitis, with approximately 68,000 cases occurring each year [105]. Reporter gene assays and quantitative RT-PCR experiments revealed that the JEV NS5 protein suppresses polyI:C-stimulated IFN-β expression [104]. NS5 expression also reduced IRF3 and p65 levels in nuclear fractions following stimulation with polyI:C [104]. Further analysis determined that NS5 expression did not impact IκBα degradation or IRF3 phosphorylation, suggesting that NS5 acts downstream of IRF3 and NFκB activation [104]. Focusing on nuclear import, co-immunoprecipitation experiments determined that NS5 interacts with IMPA1, IMPA3, and IMPA4 [104]. Moreover, this interaction impaired IRF3 or p65 binding to IMPA3 and IMPA4 in a competitive manner, as assessed by co-immunoprecipitation studies [104]. Two putative NLSs previously identified in dengue virus NS5 are conserved in JEV NS5 [104,106]. Mutational studies determined that one of the potential NLSs, found between amino acids 371–405, was crucial for IMPA binding [104]. Within this region, a ^391^KRPR^394^ motif was demonstrated to make a significant contribution to IMPA3 and IMPA4 interactions, with K391 and R394 being key residues [104]. Overexpression of IMPA3 and IMPA4 led to increased IFN-β expression and reduced JEV replication, consistent with a competitive binding mechanism of inhibition. Future studies are required to determine how the JEV NLS interacts with IMPA and whether it targets a cNLS binding site.

### 4.4. Human Immunodeficiency Virus 1 Vpr

Human Immunodeficiency Virus 1 (HIV-1) is a member of the *Retroviridae* family and causes acquired immunodeficiency syndrome (AIDS). The HIV-1 ssRNA genome encodes 9 genes, including 6 regulatory genes that have various roles in promoting HIV-1 replication, budding, and pathogenesis [107]. Virus protein r (Vpr) is one such regulatory protein that has been shown to interact with several host proteins, including human, mouse, and yeast IMPA proteins [45,108,109,110]. To determine how Vpr promotes HIV-1 replication in macrophages, investigators examined whether Vpr could suppress innate immune signaling following stimulation with different innate immune agonists. In THP-1 cells, Vpr expression suppressed expression from an IFIT-1 luciferase reporter gene following stimulation with cGAMP, herring testis (HT) DNA, Sendai virus (SeV), or LPS [45]. This suggested that Vpr can broadly suppress innate immune signaling mediated by DNA and RNA sensing pathways [45]. To more closely examine how Vpr mediates suppression of innate immune signaling, the authors incorporated different Vpr mutants into VSV-G pseudotyped HIV-1 GFP virus-like particles (VLPs). The Vpr mutant R80A, which is defective in cell cycle arrest, was still able to suppress cGAMP-induced IFIT1 promoter-driven luciferase activity in IFIT1-luciferase THP-1 cells, suggesting that Vpr-mediated cell cycle arrest is not required to suppress innate immune signaling [45]. In contrast, Vpr Q65R, which cannot recruit DCAF1 and degrade target proteins, and F34I/P35N, which fails to bind cyclophilin A and does not localize to the nuclear membrane, showed reduced suppression of the IFIT1 promoter-luciferase activity [45]. Similar results were also observed in THP-1 cells expressing a NFκB promoter-driven luciferase reporter [45]. To more closely examine the mechanism of Vpr-mediated suppression of innate immune signaling, the authors examined the phosphorylation of STING, TBK1, and IRF3 in THP-1 cells stimulated with HT DNA [45]. Vpr expression from a lentiviral vector did not alter STING, TBK1, or IRF3 S386 phosphorylation [45]. However, examination of IRF3 S396 phosphorylation by flow cytometry revealed that Vpr expression reduced IRF3 S396 phosphorylation in THP-1 cells stimulated by cGAMP or HT DNA, but the significance of this inhibition was not explored further [45]. The authors also examined whether the Vpr mutants defective in suppressing innate immune signaling exhibited altered subcellular localization. When overexpressed, Vpr localizes to the nucleus and nuclear pores [111]. However, overexpressed Vpr Q65R and F34I/P35N exhibited reduced nuclear localization and reduced nuclear rim staining [45]. Considering that Vpr mutants unable to suppress innate immune signaling also exhibited altered nuclear localization, the authors sought to determine whether Vpr could inhibit the nuclear localization of IRF3. Following differentiation of THP-1 cells with phorbol-12 myristate acteate (PMA), cells were treated with cGAMP and infected with HIV-1 GFP VLPs expressing Vpr. Compared to cells infected with VLPs lacking Vpr, Vpr expression suppressed cGAMP and HT-DNA-induced IRF3 nuclear import [45]. In contrast, VLPs with Vpr Q65R or F34I/P35N were less efficient at suppressing IRF3 nuclear localization [45]. Similar results were also observed for NKκB p65 nuclear import [45]. Through co-immunoprecipitation experiments, the authors confirmed that Vpr interacts with IMPA1, IMPA4, and IMPA5, and this interaction was reduced for the Vpr F34I/P35N mutant [45]. Furthermore, the authors confirmed that infection of cells with an HIV-1 vector encoding Vpr does not lead to the degradation of any of the KPNA proteins, as determined by Western blot [45]. Instead, Vpr expression led to reduced IMPA5-IRF3 and IMPA5-p65 interaction in co-immunoprecipitation experiments, suggesting that Vpr binds IMPA to prevent IRF3 and NKκB interaction [45]. In support of this, the Vpr mutant F34I/P35N, which is defective in suppressing innate immune signaling, did not disrupt the IMPA5-IRF3 or IMPA5-p65 interaction [45]. Given that Vpr was shown to bind representative IMPA proteins from each of the IMPA subfamilies, this suggests that Vpr may broadly disrupt the nuclear import of other IMPA cargo.

### 4.5. African Swine Fever Virus MGF360-12L

African swine fever virus (ASFV) causes African swine fever, an economically important disease of pigs characterized by fever, hemorrhage, and ataxia. The virus is classified as belonging to the *Asfarviridae* family. It possesses a 170 kb to 190 kb dsDNA genome and replicates in the cytoplasm of cells. The virus encodes more than 160 proteins, of which a number are devoted to innate immune evasion [112]. Among these is the 12L protein of multigene family (MGF) 360 (MGF360-12L), which blocks NFκB activation via competition with IMPAs [46]. MGF360-12L was demonstrated to inhibit polyI:C-induced activation of the IFNβ promoter and to block polyI:C-triggered expression of endogenous innate immune genes. MGF360-12L also inhibited the nuclear accumulation of NFκB p65 and p50 induced by TNFα, as assessed by both cell fractionation and immunofluorescence microscopy. MGF360-12L expression did not prevent IkBα degradation following TNFα treatment, suggesting a downstream step was affected to prevent nuclear localization. In support of a competition-binding mechanism, MGF360-12L interacted with IMPA1, IMPA3, and IMPA4 by co-immunoprecipitation. Further, MGF360-12L expression decreased the efficiency with which p65 co-precipitated with IMPA1, 3, and 4, and over-expression of these IMPAs overcame the MGF360-12L-mediated inhibition of IFNβ expression. Interestingly, MGF360-12L expression also blocked the nuclear import of GFP fused to a classical NLS [46]. This suggests that MGF360-12L may have broad impacts on the nuclear import of innate immune transcription factors as well as other cargo with classical NLSs.

### 4.6. Vaccinia Virus A55 Protein

Vaccinia virus (VACV) belongs to the *Orthopoxvirus* genus of the Poxvirus family, a family that also includes the now extinct smallpox virus and Mpox (formerly monkeypox) virus [113]. Like ASFV, VACV is a double-stranded DNA virus with a large genome that replicates in the cell cytoplasm. Many immune-modulatory genes have been identified in VACV [114]. One immune modulator is the A55 protein, a member of the BBK (broad complex, tram-trac, and bric-a-brac [BTB] and C-terminal Kelch) family [47]. BTB-Kelch proteins recruit Cullin 3-based E3 ligase complexes to substrates [115]. A55 expression inhibited NFκB activity in response to both IL-1β and TNF-α [47]. The effects were specific to NFκB, as A55 did not inhibit JAK-STAT-dependent or AP-1-dependent gene expression. The inhibitory activity was demonstrated to occur downstream of IκBα degradation. It was also noted that A55 expression prevented phosphorylation of p65 at residue S276, which occurs in the nucleus. When p65 nuclear translocation was assessed, A55 expression blocked nuclear localization in sixty-two percent of cells. Immunoprecipitation assays identified an interaction with IMPA1 but not IMPA4 or IMPA5. The A55 Kelch domain was demonstrated to be sufficient to interact with IMPA1 and to impair p65 nuclear import following TNFα stimulation [47]. These data indicate that A55 can prevent p65-IMPA1 interaction and suggest that this contributes to inhibition of NFκB.

## 5. Conclusions

The ability of viral proteins to competitively inhibit the interaction between transcription factors and IMPA is one of many possible viral strategies used to counteract host antiviral responses. In the case of EBOV VP24 and MERS-CoV ORF4b, structural studies have provided a mechanistic understanding as to how these viral proteins compete with antiviral transcription factors to bind IMPA. In contrast, less is known about the binding modes of other viral proteins that can compete for IMPA binding. Future studies examining these interactions can provide detailed information about the mechanisms that determine the interaction between viral proteins and IMPA. This information may also give clues regarding other cellular processes that may be impacted by these interactions, especially considering that an interaction with IMPA can potentially hinder the binding of other IMPA cargo. The interaction between viral proteins and IMPA also suggests that these viral proteins could potentially traffic to the nucleus via IMPA. Indeed, a nuclear export signal for EBOV VP24 has been identified, demonstrating that VP24 can also undergo nuclear and cytoplasmic trafficking [116]. Further work is required to understand the functional significance of this trafficking during viral replication. The studies described above provide valuable insight into how viruses can subvert the crucial role of IMPA to promote efficient viral replication, helping to provide a more thorough understanding of viral innate immune evasion strategies.

## Figures and Tables

**Figure 1 cells-13-00071-f001:**
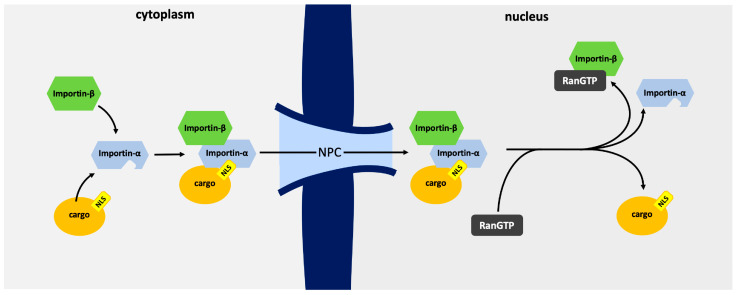
Overview of the Canonical Nuclear Import Mechanism. The nuclear cargo (highlighted in orange) harbors a nuclear localization signal (NLS) (depicted in yellow). The NLS is bound by importin-α (colored blue), and importin-β (shown in green) binds importin-α through the IBB domain. The trimeric assembly translocates through the nuclear pore complex into the nucleus, where upon entry, RanGTP (colored black) binds importin-β, mediating disassembly of the complex and the release of cargo.

**Figure 2 cells-13-00071-f002:**
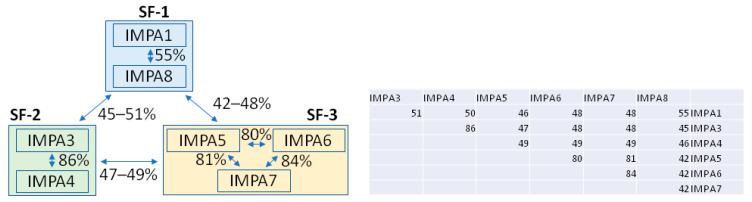
Classification of Human Importin Alpha Isoforms The seven human importin alpha isoforms are grouped into three subfamilies (SF-1, SF-2, and SF-3) according to their sequence similarities. Subfamily SF-1 includes IMPA1 and IMPA8, which exhibit the lowest sequence identity among the subfamilies. Subfamily SF-2, with the highest sequence homology, consists of IMPA3 and IMPA4. Subfamily SF-3 comprises IMPA5, IMPA6, and IMPA7. The specific sequence identity percentages, calculated using BLAST, are indicated for each isoform to the right of the figure.

**Figure 3 cells-13-00071-f003:**
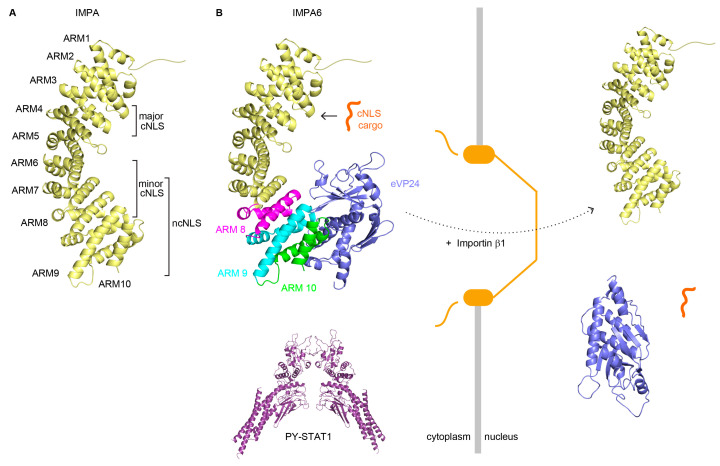
Ebola VP24 inhibits STAT1 signaling by binding SF-3 IMPAs. (**A**) IMPAs are comprised of 10 armadillo (ARM) repeats of 3 helices (PDB 1BK5). cNLS and ncNLS sites are indicated. (**B**) VP24 functions as a competitive inhibitor of STAT1-IMPA interaction by blocking phosphorylated STAT1 (PY-STAT1) binding via its ncNLS to IMPA (PDB 1BK5, 4U2X, and 1BF5).

**Figure 4 cells-13-00071-f004:**
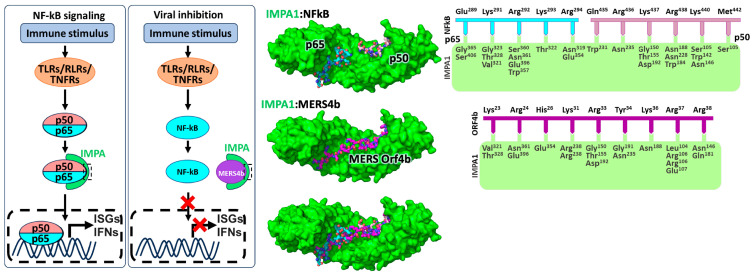
Interactions and Structural Insights into MERS-CoV ORF4b Inhibition of Immune Signaling. Normally, an immune stimulus would trigger the NF-B pathway, leading to the transport of p50/p65 into the nucleus. The complex of IMPA2 with p50/p65, which facilitates this process, has been structurally characterized (PDB 7LET). The MERS-CoV ORF4b protein can disrupt this interaction, thereby preventing the nuclear import of p50/p65 and suppressing the expression of ISGs and IFNs. Structural analysis has provided insights into the MERS-CoV ORF4b mode of action by revealing a mutually exclusive binding mechanism at the interface, as shown in the structural data from PDB 7RFZ [44]. In the visualization, IMPA is rendered in green with a surface representation, while p50 and p65 are depicted in stick form in salmon and light blue, respectively. These models are constructed based on high-resolution X-ray crystallography data.

**Table 1 cells-13-00071-t001:** Virus-encoded proteins that target IMPA block the nuclear import of innate immune transcription factors.

Viral Protein	IMPA Target	Transcription Factor Affected	Ref.
Ebola virus (EBOV) VP24	IMPA5, 6, 7	STAT1	[41]
Hepatitis B virus (HBV) polymerase	IMPA5	STAT1	[42]
SARS coronavirus (SARS-CoV) ORF6	IMPA1	STAT1	[43]
Middle East respiratory syndrome coronavirus (MERS CoV) Orf4b	IMPA3, 4, others	NFκB p50/p65	[26,44]
Hendra virus (HeV) and Nipah virus (NiV) W	all IMPAs	NFκB p50/p65	[15]
Human immunodeficiency virus 1 Vpr	IMPA1, 4, 5	IRF3, NFκB p50/p65	[45]
African swine fever virus (ASFV) MGF360-12L	IMPA3?	NFκB signaling	[46]
Vaccinia virus A55	IMPA1	NFκB	[47]

## Data Availability

Not applicable.

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
