# Peer review of "Viral Targeting of Importin Alpha-Mediated Nuclear Import to Block Innate Immunity"

_cells, 2023, doi:10.3390/cells13010071_

Round 1
Reviewer 1 Report
Comments and Suggestions for Authors
The manuscript by Vogel et al provides a detailed overview of the literature on viral mechanisms of immune evasion that target importin-alpha (IMPA) to impede nuclear translocation of key host transcription factors. It is very well written and thoroughly cites the pertinent sources. The manuscript will be of broad interest to researchers studying viral inhibition of the type I interferon and innate immune responses. I have only a few minor suggestions and comments as outlined below.
Line 64: I appreciate the effort made to explain why IMPA2 is missing, but still found this explanation unclear.
Line 106- ‘of NFKB’?
In the overview of IFN signaling include mention of ISGF3 (STAT1/2 and IRF9) as the complex that is ultimately imported into the nucleus.
Line 171- The text following the end of this line is missing.
Line 202- define NPI-1
Author Response
We thank the reviewer for the useful comments. Please see our attached responses.

Reviewer 2 Report
Comments and Suggestions for Authors
see attached file

Author Response

(The authors gave the same response as above.)
